# Updates on the Regulatory Framework of Edited Organisms in Brazil: A Molecular Revolution in Brazilian Agribusiness

**DOI:** 10.3390/genes16050553

**Published:** 2025-04-30

**Authors:** Nicolau B. da Cunha, Jaim J. da Silva Junior, Amanda M. M. Araújo, Ludmila R. de Souza, Michel L. Leite, Gabriel da S. Medina, Gustavo R. Rodriguez, Renan M. dos Anjos, Júlio C. M. Rodrigues, Fabrício F. Costa, Simoni C. Dias, Elíbio L. Rech, Giovanni R. Vianna

**Affiliations:** 1Faculty of Agronomy and Veterinary Medicine (FAV), Campus Darcy Ribeiro, University of Brasilia (UnB), Brasília 70910-970, DF, Brazil; jaimjunior@gmail.com (J.J.d.S.J.); gabriel.medina@unb.br (G.d.S.M.); 2Post-Graduation in Agronomy Program (PPGA), Campus Darcy Ribeiro, University of Brasilia (UnB), Brasília 70910-970, DF, Brazil; amandamedrado.engagro@gmail.com (A.M.M.A.); ludmilaraulino@gmail.com (L.R.d.S.); 3Genomic Sciences and Biotechnology Program, Catholic University of Brasilia, Brasília 71966-700, DF, Brazil; michelleitte@gmail.com (M.L.L.); simoni@p.ucb.br (S.C.D.); 4Agribusiness Management Course, Campus Darcy Ribeiro, University of Brasilia (UnB), Brasília 70910-970, DF, Brazil; 5Post-Graduation in Molecular Biology Program (PPGMol), Campus Darcy Ribeiro, University of Brasilia (UnB), Brasília 70910-900, DF, Brazil; r.miguel4778@gmail.com; 6Instituto de Investigaciones en Ciencias Agrarias de Rosario, Consejo Nacional de Investigaciones Científicas y Técnicas, Universidad Nacional de Rosario (IICAR-CONICET-UNR), Campo Experimental Villarino, Zavalla S2125ZAA, Argentina; grodrig@unr.edu.ar; 7Facultad de Ciencias Agrarias, Universidad Nacional de Rosario, Parque Villarino, CC Nº 14, Zavalla S2125ZAA, Argentina; 8Embrapa Genetic Resources and Biotechnology, PqEB W5 Norte, Brasília 70770-917, DF, Brazil; julio.carlyle@embrapa.br (J.C.M.R.); elibio.rech@embrapa.br (E.L.R.); giovanni.vianna@embrapa.br (G.R.V.); 9Cancer Biology and Epigenomics Program, Northwestern University’s Feinberg School of Medicine, Chicago, IL 60611, USA; fcosta@genomicenterprise.com; 10AIx4ALL, San Francisco Bay Area, CA 94110, USA

**Keywords:** edited organisms, genome editing technologies, biotechnological products, regulatory framework

## Abstract

Genome editing technologies have revolutionized the production of microorganisms, plants, and animals with phenotypes of interest to agriculture. Editing previously sequenced genomes allows for the punctual, discreet, precise, and accurate alteration of DNA for genetic analysis, genotyping, and phenotyping, as well as the production of edited organisms for academic and industrial purposes, among many other objectives. In this context, genome editing technologies have been causing a revolution in Brazilian agriculture. Thanks to the publication of Normative Resolution No. 16 (in Portuguese Resolução Normativa No. 16-RN16) in 2018, Brazilian regulatory authorities have adapted to the new genetic manipulation technologies available to the scientific community. This review aims to describe the effects of updates to the regulatory framework for edited organisms in Brazil and to point out their impacts on research and development of emerging technologies in the Brazilian agricultural sector. The implementation of RN16 rationalized the regulatory aspects regarding the production, manipulation, exploration and commercial release of edited organisms and led to the faster, cheaper and safer obtaining of edited technologies, which are more productive and better adapted to different environmental conditions in Brazil.

## 1. Introduction

Agriculture and livestock farming are human activities that were essential for the transition from nomadic tribal social organizations to complex and sophisticated civilizations. From approximately 10,000 BC, the domestication of cereals such as wheat, barley and rye, as well as dicotyledons such as flax, rapeseed and potatoes, carried out by peoples of Classical Antiquity, such as the Sumerians and Akkadians, represented the first technological revolution capable of improving the qualitative and quantitative conditions regarding the production of food, fiber and energy [1,2].

After centuries of rudimentary and incipient improvement, based on empirical crossings and selection of plants and animals, the rediscovery of Mendel’s Laws at the beginning of the 20th century allowed the understanding of the mechanisms that govern heredity and their scientific application for conventional/traditional plant and animal genetic improvement from the 1950s onwards [3].

Since then, a wide variety of annual and perennial plants, as well as animals, have been the target of conventional breeding aimed at obtaining varieties and individuals that are tolerant/resistant to forms of biotic and abiotic stress, in addition to maximizing productivity, reducing production costs and making management easier [3].

In the early 1970s, recombinant DNA technology allowed the cloning and expression of exogenous genes in bacteria, opening a new frontier in research involving molecular genetics: the in vitro genetic manipulation of microorganisms, plants and animals for the purposes of increasing agricultural productivity, nutritional enrichment, biosynthesis of fibers and molecules of industrial and therapeutic interest [4]. New traits and characteristics were obtained by the direct introduction of genes of interest, carried by specific gene constructs (recombinant DNA vectors), into the genome of host species, obtaining the first genetically modified organisms (GMOs) [5,6]. Several genetic transformation methods have been developed for the purpose of obtaining GMOs, such as the indirect method that uses the soil bacterium *Agrobacterium tumefaciens* as a biological agent mediating the insertion of exogenous genetic material into plant cells, in addition to physical and chemical methods of direct introduction of genes such as microparticle bombardment (Biolistics), electroporation and chemical transfection of protoplasts [7,8].

The revolution in agribusiness caused by GMOs has been unprecedented. Over the past two decades, the global area cultivated with genetically modified (GM) crops has grown a thousandfold, totaling more than 206.3 Mha in 2023 [9,10,11]. Among the main countries producing agricultural products, Brazil ranks second in the international ranking of area planted with GM crops (66.9 Mha), behind only the United States (74.4 Mha) [10,11].

In addition to the Flavr Savr^TM^ tomato with longer shelf life, the first transgenic plants to reach the American market, such as Round-up Ready^TM^ (RR) soybeans and Bt cotton, allowed, respectively, greater ease of weed management and insect pest control, with a significant reduction in the use of pesticides and increased profitability for rural producers [12,13]. These advantages explain the success of the massive adoption of GM crops in the world. In Brazil, in the 2018/2019 harvest, the percentages of transgenic plant planting broken down by the three main crops were 95% for soybeans, 88% for corn (first and second crop seasons) and 84% for cotton [10].

However, obtaining GM plants is not a trivial, quick and cheap process [14]. Several reasons explain the difficulties in obtaining elite engineered plants, with desirable phenotypes associated with high levels of expression of the gene(s) of interest:The high complexity and size of plant eukaryotic genomes, notably presenting extensions of the order of a few hundred million bases (Megabase) to a few billion bases (Gigabase) [15].The abundance of extensive non-coding genomic regions full of repetitive sequences [16].The randomness of recombination of DNA fragments carrying genes of interest in genomic sites of low gene expression, when site-directed recombination techniques are not used [17,18].The presence of extensive regions of heterochromatin associated with hypercondensed DNA, where genes often present low levels of expression [19].The position effect associated with the integration of multiple copies of the gene of interest, often associated with phenomena of post-transcriptional silencing of gene expression [7,20].

In the 1980s, molecular markers such as restriction fragment length polymorphisms (RFLPs) began to be used as tools to aid in the selection of plants and animals in genetic improvement programs [21]. The selection of individuals assisted by molecular markers contributes to greatly increasing the accuracy of the choice of genotypes for crossbreeding and to accelerating the production of elite varieties with the best phenotypes for agricultural production purposes [22].

From the 2000s onwards, with the technological advances in large-scale DNA sequencing and genomic mapping, the selection of individuals began to be based on the identification of multiple markers and alleles of interest simultaneously, with a notable increase in precision and accuracy in identifying the best genotypes [23,24].

The development of gene editing systems, notably zinc-finger nucleases (ZFNs), transcription activator-like effector nucleases (TALENs) (in the early 2000s) and, mainly, the use of the clustered regularly interspaced short palindromic repeats/CRISPR-associated protein 9 (CRISPR/Cas9) system, since 2010, have allowed the acquisition of variability to be explored in genetic improvement programs [25]. Genetically edited plants and animals, with new characteristics of tolerance/resistance to biotic and abiotic stresses, and new nutritional and energetic properties represent a new paradigm of productivity in the field [26,27].

The CRISPR/Cas9 system, the most widely used gene editing technology today, allows for the precise and discrete performance of specific modifications in previously mapped target genomic sites [28]. These modifications can be gene knock-in or knock-out, point mutations, epigenome editing, gene replacement, multiplexed genome editing, gene transcriptional regulation and chromosomal rearrangement [27].

Given the highly specific recognition and processing characteristics of the genomic target site(s) for editing by the Cas9 enzyme, mediated by a hybrid structure formed by the target DNA and the single guide RNA (sgRNA), the CRISPR/Cas9 system allows for the performance of desirable genomic editing with a low level of modification at non-target sites (off-targets). As a direct consequence, for example, an elite edited plant with the desired phenotype can be obtained in a time interval and at considerably lower costs than those associated with an elite GM plant [7].

Recently, “Speed Breeding” (SB) has been used to accelerate the development cycle of photoperiod-responsive plant varieties, such as soybeans, in genetic improvement programs [29]. SB is a tool for shortening the time in advancing generations of plants, which are grown in greenhouses with strictly controlled plant growth parameters, such as photoperiod, humidity, temperature, soil fertility and the incidence of pathogens and pests [29]. Through SB, it is possible not only to evaluate the stability of genes of interest in successive generations, but also to accelerate the development of elite conventional, GM or edited varieties [29].

Figure 1 exemplifies the main technological advances associated with the increase in plant productivity in recent millennia, as well as the technical challenges inherent in obtaining GM plants and the technological revolution caused by the main gene editing systems in obtaining edited plants.

The main objective of this review is to present the current panorama of the Brazilian regulatory framework concerning GMOs and edited organisms and the impacts that updates to the Biosafety Law are beginning to promote in the country’s biotechnology scenario.

## 2. The Brazilian Biosafety Law and GMOs

The Brazilian Biosafety Law (Law No. 11,105, 24 March 2005) is the legal instrument that established the safety standards and regulations and the monitoring and evaluation mechanisms for all activities related to GMOs and their by-products in the country [30]. The activities regulated by the law are: construction, cultivation, production, handling, transportation, transfer, import, export, storage, research, environmental release, unloading, and commercialization of GMOs and their derivatives, covering both research activities and commercial uses of technologies developed for human and animal health, agriculture, livestock, and the environment [30].

The current Brazilian Biosafety Law-regulated by Decree No. 5591 of 22 November 2005-is derived from the first Biosafety Law (No. 8974 of 1 May 1995), enacted to regulate the first GMOs used commercially in the country-notably the first GM crop in Brazil, Monsanto’s RR^TM^ glyphosate-tolerant soybean, approved for commercial use in 1998 [30]. The Biosafety Law established a National Biosafety Policy (NBP), supported by a decentralized regulatory system composed of four regulatory, inspection, analysis and technical consultation institutions/bodies, with decision-making authority supported by an intensive exchange of technical information and data on the performance, risk and safety of GMOs evaluated in vitro and in vivo [30].

The four bodies are: (a) National Biosafety Council (in Portuguese CNBS–Conselho Nacional de Biossegurança), (b) National Technical Biosafety Commission (in Portuguese CTNBio–Comissão Técnica Nacional de Biossegurança), (c) Local Biosafety Committee (in Portuguese CIBio–Comissão Interna de Biossegurança) and (d) Registration and Inspection Organizations and Entities (in Portuguese OERFs-Órgãos e Entidades de Registro e Fiscalização), which includes the Ministry of Agriculture, Livestock and Supply (in Portuguese MAPA–Ministério da Agricultura, Pecuária e Abastecimento); the Ministry of Health (in Portuguese MS–Ministério da Saúde), the Ministry of the Environment (in Portuguese MMA–Ministério do Meio-Ambiente), and the Secretariat of Aquaculture and Fisheries [30].

The composition and attributions/competencies of each of these bodies are determined by the Biosafety Law and can be summarized in Table 1 below.

The system of agencies defined by the Biosafety Law is hierarchical, with the CNB as the highest authority in the decision-making process for the commercial release of GMOs and their derivatives. However, it is also decentralized, based on the CIBios–essential for ensuring Biosafety in the country–and the CTNBio, which is sovereign in the regulation of research activities involving GMOs and plays a fundamental role in carrying out technical judgments on the commercial release of GMOs and their derivatives [31,32]. Figure 2 summarizes the organization of the institutional Biosafety system in Brazil.

## 3. Legal Regulation of Published Organizations in Brazil: Normative Resolution No. 16

Obtaining more productive plants and animals via genetic improvement programs depends on strategies for exploring population genetic variability through candidate selection and targeted crossbreeding. Variability can be innate to a segregating population or induced by various techniques for manipulating organisms in the laboratory and in the field [5]. Germplasm banks contain gene pools contained in seeds, seedlings, vegetative propagation structures, semen, ovules and cryopreserved embryos, which can be used as donor sources of alleles associated with phenotypes of interest [33]. In the case of plants, when populations of individuals present low innate genetic variability, it is possible to obtain new alleles and multiple chromosomes sets with new genomic rearrangements by exposing populations to mutagenic agents and through nuclear fusion of plant cells devoid of cell walls (protoplasts), among other methods [33].

Another possibility for obtaining new plant genotypes is through the development of GM crops. There are two types of GM crops: transgenic plants, which undergo genetic modification in vitro by adding genes of interest donated by species with which the recipient plant is not sexually compatible; and cisgenic plants, whose new genes originate from species capable of crossing with the recipient plant [7,34].

A third approach is the use of emerging technologies, called Innovative Precision Improvement Techniques (in Portuguese TIMPs –Técnicas Inovadoras de Melhoramento de Precisão), also known as New Breeding Technologies (NBTs) –among which are genome editing systems–for the purpose of acquiring genetic variability. Edited plants accumulating specific genomic alterations may be capable of crossing with previously developed cultivars to introgress genes encoding desirable characteristics into progenies of high commercial value [32].

To avoid creating obstacles to technological advancement and at the same time ensure biosafety parameters, updates and adaptations to legislation need to be made to accommodate the effects caused by emerging technologies and to make safety requirements proportional to the risks.

Many of the NBTs currently routinely used in research center laboratories were incipient or simply did not exist in the 1990s, when the Biosafety Law was enacted, which reinforced the need for adjustments to the legislation.

Thus, in 2015, CTNBio organized a committee of members to technically evaluate the products obtained using NBTs and determine how they fit into the terms of the Biosafety Law, in addition to suggesting the creation of accessory legislation that covers cases not yet regulated by the legal text. Several types of NBTs were evaluated, such as genome editing, reverse breeding, RNAi (post-transcriptional gene silencing by interfering RNA), directed mutagenesis and early flowering [32,35].

The vast majority of NBTs were considered methods of accelerating the pipeline for obtaining plants and animals in breeding programs, caused by the rapid introduction of new traits in elite genotypes. Depending on the molecular approach of modification caused by the NBT, the final organism obtained and/or its derivatives could be legally classified as non-genetically modified (non-GMOs) [32,35].

Regarding the use of the CRISPR/Cas9 system, one of the main NBTs, the genetic modifications caused in the genome of plants and other organisms are discrete and punctual (often restricted to a change in a single nucleotide). Frequently, the edited genomic site(s) present extensions in base pairs (bp) significantly smaller than the total size of the genome of the target organism, characterizing the editing, from a structural and functional point of view, equivalent to an induced mutation. In addition, at the sequence level, edited mutants are indistinguishable from spontaneous mutations or those induced by chemicals or radiation and the editing process leaves no other footprints behind [5,12,36].

When a genome editing system is used to produce mutations in an organism’s genome, the mutation generation strategy is said to use site-directed nucleases (SDNs). There are three approaches to using SDNs: (SDN-1, SDN-2, and SDN-3) [32].

In SDN-1, the DSB caused by ZFNs, TALENS, or the CRISPR/Cas9 system is repaired by an endogenous NHEJ-type DNA repair system, leading to simple, random mutations that often silence the gene [7,32].

In SDN-2, after the DSB occurs, a DNA template fragment carrying a mutation and ends with bases homologous to the edges of the lesion is recombined at the site where the DNA is cleaved, by the HDR mechanism. In this way, the original base sequence is replaced by a new, virtually identical sequence (donated by the template fragment), carrying the mutation [7,32].

The SDN-3 approach involves the use of the NHEJ or HDR repair pathways to insert into a genomic site one or more DNA fragments carrying genetic regulatory elements for gene expression, such as promoters and terminators, flanking the coding sequence of a gene of interest [7,32].

Obtaining mutants using SDNs is relatively recent. For decades, however, the use of classical mutagenesis techniques, such as bombardment with ionizing radiation and chemical mutagenesis, has allowed the production of mutants. The main difference between SDNs and classical mutagenesis methods is that the former allow the generation of point and site-specific mutations, while the latter generate random and multisite mutations [7,32].

In this context, since the Brazilian Biosafety Law considers mutant organisms obtained by classical mutagenesis techniques to be non-GMOs and since some edited organisms (obtained by SDN-1 and SDN-2) are equivalent to mutants, these could also be classified as non-GMOs. A legal impasse was created that led to the need for adjustments to the legislation [16].

Therefore, the CTNBio NBT evaluation committee studied the biosafety legislation, the broad regulatory framework and the practical experience of several leading countries in biotechnology research, and suggested an update to the legislation, through unanimous approval by the CTNBio plenary and by the Legal Counsel of the Ministry of Science, Technology, and Innovations of Normative Resolution No. 16 (RN16), in January 2018 [12,32,35].

RN16 determines that CTNBio has the authority to evaluate, on a case-by-case basis, biotechnological products obtained with the use of NBTs as GMO or non-GMO. In the evaluation, CTNBio members have access to information collected by the institution developing the product candidate, such as detailed molecular analysis and production methodology [37]. The criteria for the final classification of the product are shown in Table 2.

RN16 applies to all types of organisms, including bacteria, protozoa, yeasts, filamentous fungi, algae, plants and animals, in the research setting and in the pipeline for commercial release [37].

## 4. The GMOs and Edited Organisms’ Regulatory System in Brazil

In the case of both GMOs and edited mutants, candidates to become commercial products must comply with Brazilian regulations as provided for by law, since their production depended on the in vitro genetic manipulation of plant genotypes [32].

The regulatory frameworks are based on extensive risk analysis protocols and technical/scientific requirements, designed to protect human and animal health, as well as the environment, from potential harm/adverse effects related to GMOs and edited organisms.

In general, the regulatory frameworks for GMOs and edited organisms differ from country to country [12,32,38,39]. There are four categories of countries based on the level of regulatory restrictions regarding the commercial release of edited organisms: (1) countries in which the regulatory framework is identical for edited organisms and GMOs (countries with more restrictive regulatory frameworks); (2) countries in which the regulatory framework for edited organisms is similar to that of GMOs but is occasionally simpler; (3) countries whose regulatory framework allows edited organisms obtained through specific approaches to be considered non-GMOs, whose regulatory framework is extremely simpler than that of GMOs, but whose commercial release depends on prior approval by a public institution/government agency; and (4) countries in which edited organisms are considered non-GMOs and do not depend on government approval for commercial release [16,22]. With the publication of RN16 and the possibility of obtaining mutant organisms edited by SDN-1 and SDN-2, Brazilian regulatory framework positioned the country in category 3 (Figure 3).

In the Brazilian regulatory system, in addition to the regulatory institutions and their compositions and attributions, the relationships between them and the sequence of procedures from the authorization of research with GMOs, in addition to the steps aimed at the commercial release of products, complete the elements that make up research/risk assessment/monitoring. RN16 complemented the system by enabling the regulation of products obtained by NBTs [12,32,38].

The steps begin with the establishment of CIBio by a research institution that aims to work with GMOs. Through CIBio, the institution requests the CQB from CTNBio. After careful analysis by CTNBio regarding technical competence, personnel training and adaptation of facilities according to required biosafety parameters, CTNBio issues the CQB and authorizes the research institution to carry out activities with GMOs [32].

Once the research institution has the CQB, it requests the commercial release of a technology through the president of CIBio, who requests the opening of a commercial release process with CTNBio and attaches to the process a complete and detailed dossier that describes all the technical aspects of the product candidate, as well as the methodological strategy used to obtain it and all biosafety risk assessments established by Normative Resolution No. 24, of January 7, 2020 [32]. CTNBio then assesses whether the product candidate is classified as GMO or non-GMO. If the assessment certifies the technology as GMO, CTNBio evaluates the risk and prepares a technical report called Risk Assessment Dossier (RAD). If the RAD concludes that the GMO is safe and it is approved for commercial release, it is forwarded to CNBS and may be monitored by CTNBio, through a monitoring plan, in case an exemption is not requested by the research institution or is not granted by CTNBio itself [32].

On the other hand, if CTNBio classifies the product as non-GMO, it can be registered using existing procedures without the need for monitoring. An overview of how the Brazilian regulatory system for products obtained by NBTs works can be seen in Figure 4.

Monitoring of a GMO product whose release has not been rejected by the CNBS is carried out according to the monitoring plan stipulated by the Biosafety Law [15,40]. The monitoring plan provides for the possibility of monitoring at two levels. If the risks associated with the GMO are so low that they are negligible, General Monitoring (GM) is carried out. If the risks cannot be neglected, Case-Specific Monitoring (CSM) is carried out (Figure 5).

## 5. Effects of RN16 on Brazilian Agribusiness

With the update of Brazilian legislation, through the establishment of RN16, there was a stimulus to the environment for research and development of technologies obtained by NBTs in the country [32]. The local increase in the dynamism of obtaining emerging technologies can be explained by several factors: acceleration in the obtaining of new non-GMO products; reduction in the final cost of product development; increase in requests for commercial release of technologies; approval, licensing and registration of products in the market; in addition to greater diversification of products and development companies in the local Biotechnology scenario [12,32,35].

In the case of plants, the use of NBTs in obtaining elite cultivars was a leap forward in breeding technology [1,26,33]. The acceleration in the development of these products can be explained by a set of technical factors inherent to genome editing technologies and due to the changes in the Brazilian regulatory framework mentioned above.

The high precision and accuracy of DNA editing of the target plant by the CRISPR/Cas9 system is caused by extensive interactions between the sgRNA and the target genomic editing site (RNA/DNA interactions), in addition to the interactions between the Cas9 nuclease and both nucleic acids (Protein/DNA/RNA interactions) [27,41,42].

In addition, unlike ZFNs and TALENS, which require two nucleases for DSB production, a single Cas9 can promote the DSB per se. In addition, the modified version of the CRISPR/Cas9 system, called Prime Editing, can use a new Cas 9 (nCas9) with one of the two cleavage domains knocked out and carrying a reverse transcriptase domain, capable of recognizing a prime editing guide RNA (pegRNA) [27,41,43]. The pegRNA-guided nCas9 can edit a genomic site by producing a single strand break (SSB) and using the pegRNA as a template for the biosynthesis of complementary DNA (cDNA) carrying the sequences of interest, which is integrated into the edited DNA [7].

The different editing modalities by CRISPR/Cas9 and Prime Editing reduce off-targets and increase the probability of obtaining individuals whose indels and/or gene insertions promote top phenotypes of interest, with the modifications stabilized/fixed in the genome and capable of being inherited by the progeny [44,45]. In contrast, mutations induced by ionizing radiation and chemical agents are usually hemizygous and to become homozygous and be fixed in the progeny depend on the observation of segregating branches and the performance of the required number of necessary backcrosses [33,46,47].

Another advantage of the CRISPR/Cas9 system is the ability of the editing system to target multiple copies of a gene, including homologous genes, in polyploid organisms [27,29]. In this way, obtaining products engineered by NBTs greatly shortens the pipeline for the development of elite cultivars of annual plants and, especially, of perennial plants with long or very long cycles, such as sugarcane, coffee, eucalyptus, poplar, citrus and other fruit trees [33,48]. The differences in the timelines for selecting genotypes of interest in fruit trees, involving different approaches to obtaining variability, selection and crossings, can be seen in Figure 6.

In the seven years following the publication of RN16, there was an increase in the entry of companies developing emerging technologies into the national market and a boom in requests for commercial release of edited products [32]. By November 2024, approximately 63 requests for commercial release of edited organisms had been requested by CIBios from 26 different biotechnology companies to CTNBio [23]. Approximately 62% of the product candidates evaluated by CTNBio (39 technologies) are microorganisms or their components/derivatives [23] (Figure 7).

For comparison purposes, from the commercial release of RR soybeans in Brazil in 1998 to 2024 (26 years), 155 GM crop events were released in the country [49].

The new environment for research and development of edited technologies is directly related to changes in Brazilian regulations. Since the steps to be completed in the GMO regulatory system are expensive, only a few large multinational companies (BASF, Bayer, Corteva, and Syngenta) and large national companies (Sempre Ag Tech) have the economic strength to fund the development of many GM crops [12,32]. Public and private universities, small and medium-sized companies, and companies funded by the Union, such as the Brazilian Agricultural Research Corporation (in Portuguese EMBRAPA–Empresa Brasileira de Pesquisa Agropecuária) are not capable of developing a wide range of GM crops with commercial product status on their own and are left out of the competition for market niches [12,32].

## 6. Conclusions

After 25 years of adoption of GMOs in agribusiness and in the use of therapeutics such as recombinant insulin in Brazil, the absence of significant impacts on human, animal and environmental health, and the advent of new emerging technologies such as genome editing, has led to the need to update the regulatory framework. The positive medium-term impacts caused by changes in legislation to accommodate edited organisms in the national Biosafety legal framework are already being felt by the local scientific community [32].

The legal certainty implemented in the country by the Biosafety Law and RN16 stimulated scientific research with transgenic and cisgenic organisms and made it possible to increase the safety of human and animal health and environmental protection.

As a direct consequence of the simplification and rationalization of the regulatory framework for emerging technologies, there was an acceleration in the development of candidate technologies for products and an increase in the flow of requests for commercial release, technical evaluation and arrival on the market of new technologies, with the potential to benefit a broader range of stakeholders, particularly in poor regions [50]. Indirectly, there was the emergence and expansion of markets associated with biosafety and biotechnology, capable of generating more jobs, income and technological advancement in the field.

## Figures and Tables

**Figure 1 genes-16-00553-f001:**
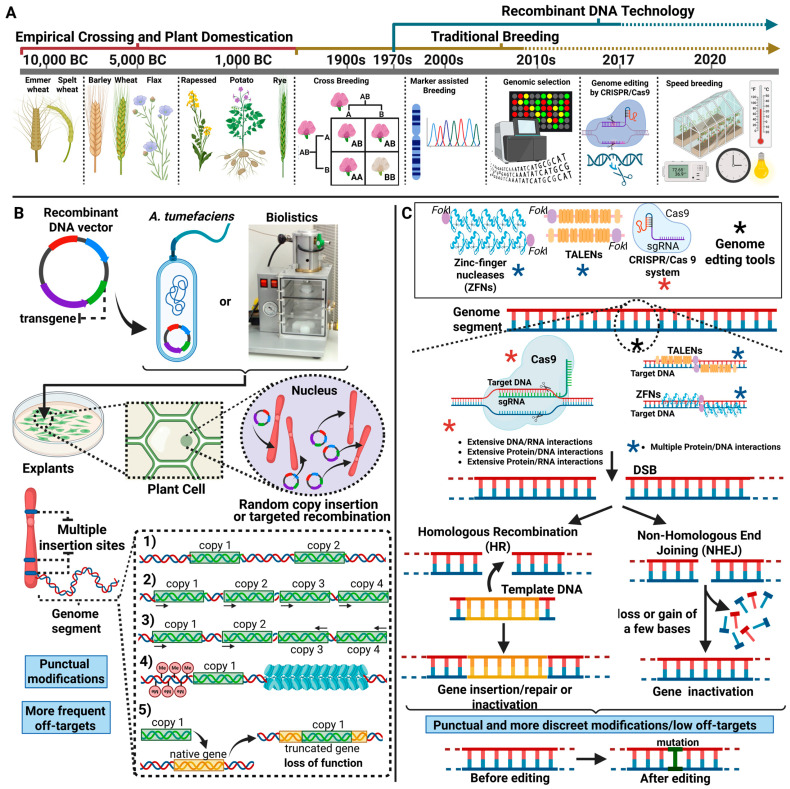
(**A**): Timeline of main advances in technology for producing more productive plants for agriculture. (**B**): After cloning of transgene in DNA vector, genetic transformation methods, such as indirect transformation mediated by *A. tumefaciens* and direct transformation by Biolistics can be used to obtain GM plants. The in vitro introduction of genes into plant explant cells by both methods can be random or mediated by site-directed recombination. In the first case, insertions of transgene copies into multiple genomic sites frequently occur, which can result in one or more phenomena such as: (**1**) low number of integrated copies in a locus of high gene expression; (**2**) transcriptional gene silencing by integration of multiple copies of the transgene in tandem; (**3**) post-transcriptional silencing resulting from integration of multiple copies in inverted orientations; (**4**) recombination of the transgene in a heterochromatin region associated with low expression levels; and (**5**) knockout of a native plant gene. (**C**): Use of genome editing technologies to increase the precision of insertion of transgene copies into previously mapped sites of high gene expression or to induce indel-type mutations. ZFNs, TALENs and the CRISPR/Cas9 system are based on the biosynthesis of specialized nucleases capable of catalyzing the production of a severe lesion at the target genomic site characterized by the cleavage of the two DNA strands, called a double strand break (DSB). After the production of the lesion, two DNA repair mechanisms can be activated: direct homologous recombination (HDR)–when a DNA fragment with complementary and homologous ends to the edges of the DSB is offered for recombination at the editing site, and the non-homologous end joining (NHEJ) pathway, responsible for causing mutations by deletion of bases for the restoration of continuous strands at the DSB site. In both cases, editing is represented by discrete modifications at the genomic level and a low level of off-targets. Created with BioRender.com (https://www.biorender.com/).

**Figure 2 genes-16-00553-f002:**
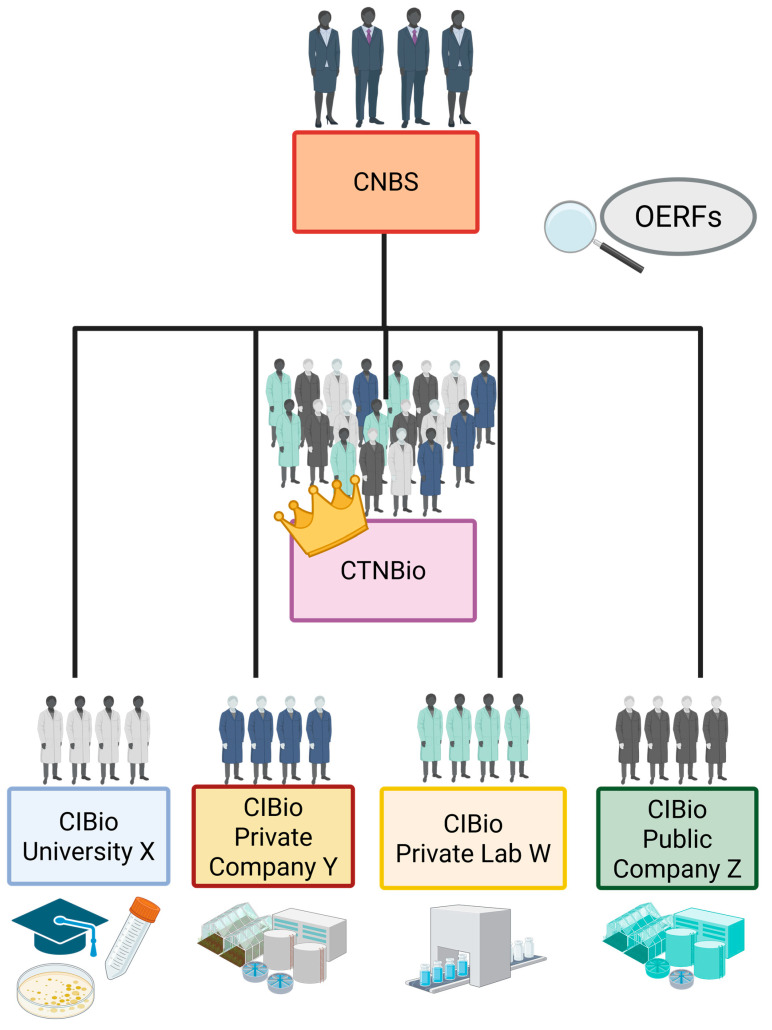
Institutions that form the hierarchical and decentralized Brazilian Biosafety system. CTNBio stands out as the most important technical body regarding the regulation of research activities with GMOs and their derivatives and the technical authority in the judgment, on a case-by-case basis, of requests for commercial release of GMOs in the country. Regarding the monitoring of GMO products, the role of OERFs stands out in monitoring and detecting possible harmful effects of GMOs on human, animal, plant and environmental health. Created with BioRender.com.

**Figure 3 genes-16-00553-f003:**
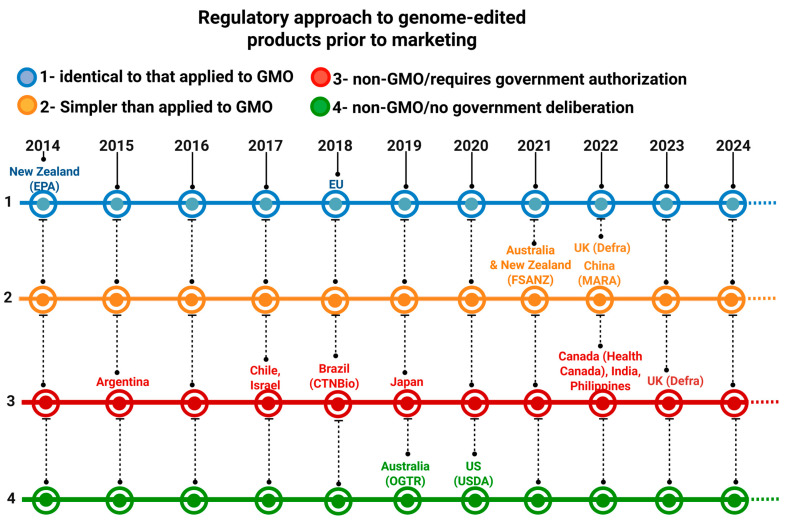
Classification of countries by regulatory approaches to the commercial release of edited organisms. Since 2014, when the first edited organisms regulatory frameworks were established, some countries have changed their legislation as a result of years of practical experience regarding the perceived risks of using edited organisms. In parentheses are the acronyms of the biosafety regulatory agencies of each mentioned country. Environmental Protection Authority (EPA); Food Standards Australia New Zealand (FSANZ); Department for Environment, Food & Rural Affairs (Defra); Ministry of Agriculture and Rural Affairs (MARA); Office of the Gene Technology Regulator (OGTR) and United States Department of Agriculture (USDA). Created with BioRender.com.

**Figure 4 genes-16-00553-f004:**
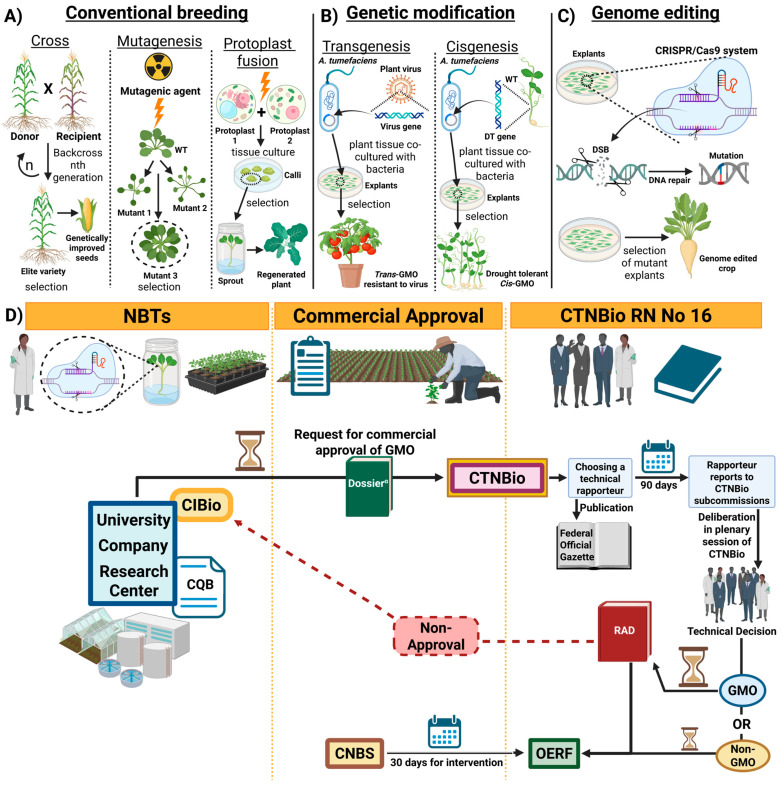
Strategies for obtaining plant variability to be explored in plant breeding programs: (**A**) Spontaneous mutants can be selected for crosses to obtain more productive individuals. Induced mutations can produce new alleles of interest for genetic improvement. Protoplast fusion contributes to increasing chromosomal variability in polyploid cells. (**B**) Transgenic or cisgenic GM plants with new phenotypes of interest are widely used in Brazilian agriculture. (**C**) The CRISPR/Cas9 editing system is the main genome editing technology. Depending on the editing approach, the final product obtained can be classified as GMO or non-GMO. (**D**) The set of steps of the process of approval and commercial release of products generated by NBTs provided for in the Brazilian regulatory framework. Created with BioRender.com.

**Figure 5 genes-16-00553-f005:**
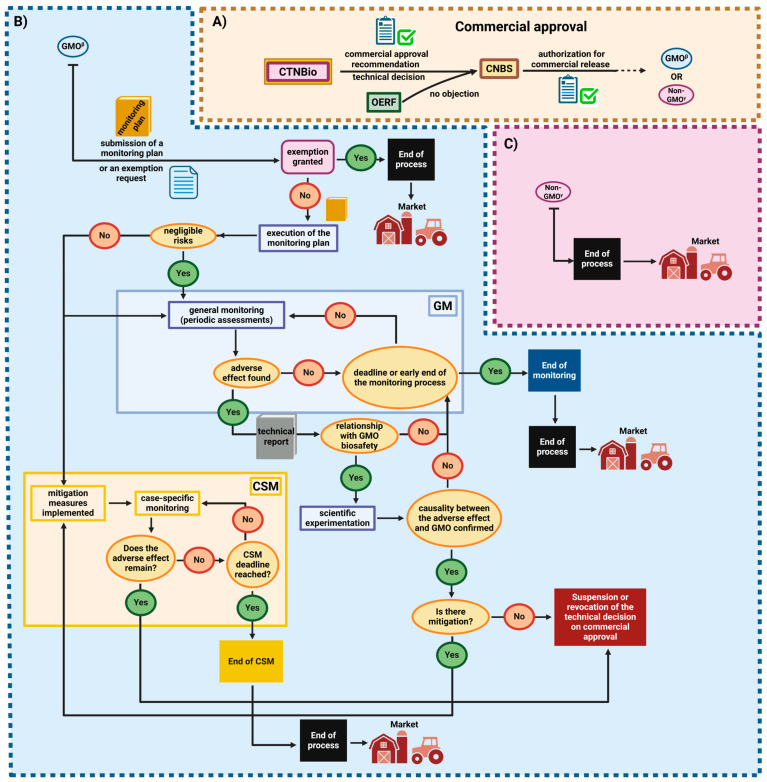
Regulatory process after commercial approval of GMO and non-GMO technologies carried out by CTNBio. (**A**) After the technology has been classified and approved for commercial release, if there is no objection by the CNBS or the OERFs, the product may follow two procedures, depending on its legal nature. (**B**) If the technology is a GMO and an exemption request is denied, it will be submitted to the monitoring plan. Depending on the results observed, the product may be released for the market or commercial approval may be suspended. (**C**) If the technology is a non-GMO, it may already be licensed and commercialized without the need for monitoring. Adapted from CTNBio [40]. Created with BioRender.com.

**Figure 6 genes-16-00553-f006:**
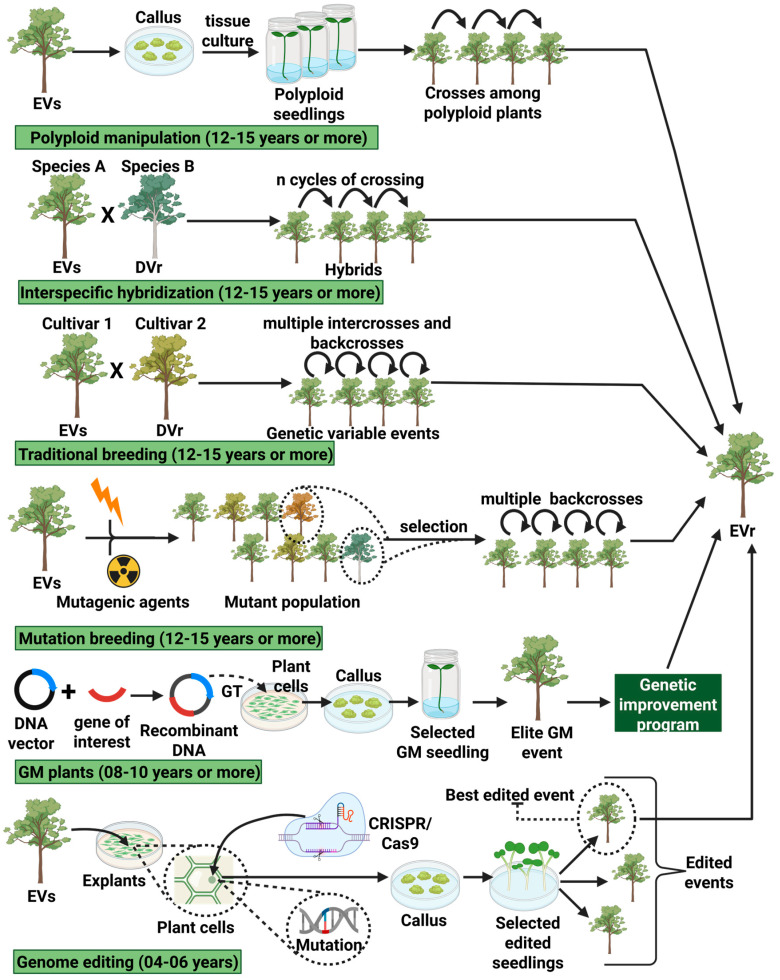
Comparison of the development timelines of fruit crops with phenotypes of interest, based on different approaches to obtaining and exploring variability. In the example, an elite susceptible variety (EVs) is a fruit plant that presents a broad phenotype with desirable characteristics for agriculture but is susceptible to a specific form of stress. A resistant donor variety (DVr) is a plant carrying a specific phenotype of interest such as resistance to stress that affects EVs. An elite resistant variety (EVr) is the plant resulting from the breeding process carrying a stress resistance phenotype. If a GM plant undergoes genome editing, the resulting cultivar will be considered NBT if chromosomal recombination and transgene/cisgene segregation have occurred, so that the plant is free from the presence of previous in vitro modifications. Gene transfer (GT). Adapted from Alvarez et al., [33]. Created with BioRender.com.

**Figure 7 genes-16-00553-f007:**
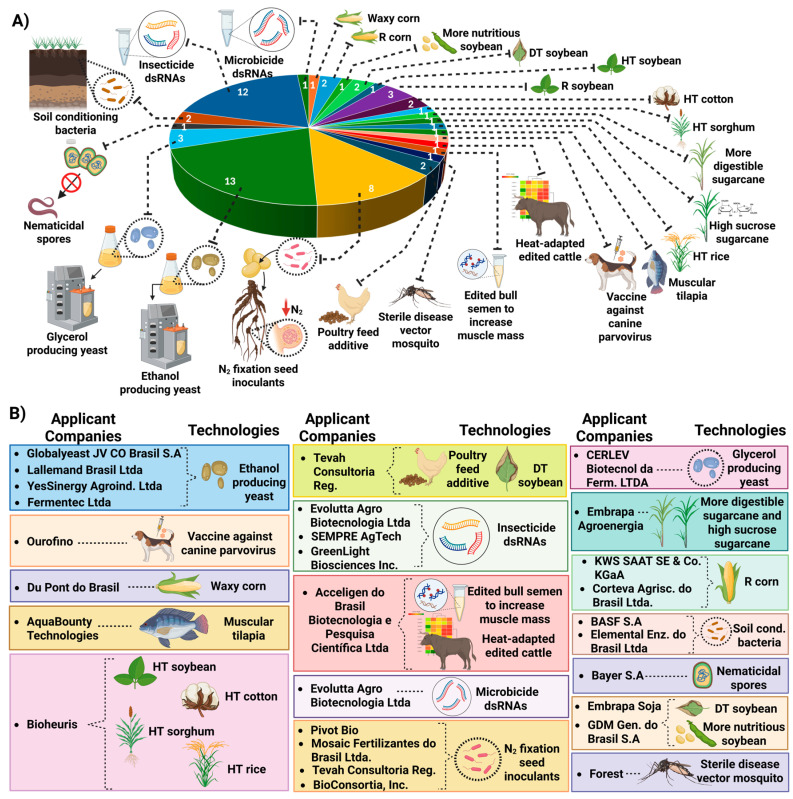
(**A**) Technologies obtained with NBTs whose commercial release requests were submitted to CTNBio between 2018 and 2024. (**B**) Technologies developed by NBTs listed by their respective developer companies [23]. R insect resistant; HT herbicide tolerant; DT drought tolerant; dsRNA double-stranded RNA. Created with BioRender.com.

**Table 1 genes-16-00553-t001:** Composition and functions of the bodies that form the Brazilian regulatory system regarding biosafety involving GMOs.

Organ	Composition	Members	Characteristics and Attributions
CNBS	11 high representatives of the Brazilian State	Chief of Staff of the Presidency (Council President) Minister of Justice Minister of Science, Technology and Innovations Minister of Agricultural Development Minister of Agriculture, Livestock and Supply Minister of Health Minister of the Environment Minister of Development, Industry and Foreign Trade Minister of Foreign Affairs Minister of Defense Secretary of Aquaculture and Fisheries	Highest authority in the Brazilian regulatory system Advises the President of the Republic on the formulation and implementation of NBP Establishes principles and guidelines for action on issues of commercial use of GMOs and their derivatives, based on issues of national, socioeconomic and political interest It speaks out whenever it wishes (quorum of 6 or more members), when requested by CTNBio or regulatory bodies If socio-economic and/or strategic policy decisions are required, the CNBS may issue a technical opinion (not technical judgment) in a final decision on the commercial release of GMOs Issues a final decision regarding the commercial release of GMOs and their derivatives. It has the power to refute, within 30 days, a technical decision recommending the commercial release of a GMO or derivative, issued by CTNBio. If it does not issue a refutation during the forementioned period, the GMO is automatically qualified for commercial registration
CTNBio	27 full members and their alternates (all with doctorates), all Brazilian citizens The 12 scientific members are directly appointed by the MCTI minister The remaining members are recommended by the Ministries that form the CNB and appointed by the MCTI minister Members must present recognized technical competence and outstanding scientific backgrounds All members must be professionally active in the areas of Biology, Biotechnology, Biosafety, Microbiology, Health and the environment, Human/animal health, or related areas	12 scientists: 3 experts in human health, 3 in zoo-health, 3 in plant health, 3 in the environment 9 representatives of ministries 1 expert in consumer protection 1 expert in health 1 expert in the environment 1 expert in biotechnology 1 expert in family farming 1 expert in worker health	Presents 4 permanent sectoral subcommittee: environment, plant health, animal health and human health Support the Federal Government in the establishment of the NBP Issue normative resolutions and biosafety guidelines for risk assessment of GMOs and their derivatives Establishes technical safety standards regarding the authorization of activities related to research, and the commercial release of GMOs Establishes standards for research with GMOs and derivatives Carry out, on a case-by-case basis, a risk assessment of activities and projects with GMOs and derivatives Authorize, register and monitor research activities with GMOs or GMO derivatives Provide technical assistance to research organizations Inspect research organizations Authorize the import of GMOs and their derivatives for research activities Advise the CNBS in formulating the NBP Issue Biosafety Quality Certificates (in Portuguese CQBs – Quality Certificates in Biosafety), an official authorization for the research institution to operate with Recombinant DNA Technology methods and be under government control Assessment of new technologies and their possible impacts on human and animal health and the environment Propose new regulations for emerging technologies Evaluate risks associated with activities with GMOs and prepare a technical report on commercial release Issue a technical decision/judgment on the biosafety and commercial release of GMOs and their derivatives, which is binding on other bodies and entities of the Public Administration Minimum quorum of 14 members at meetings, including at least one representative from each of the 4 subcommittees The committee may invite members of the scientific community, representatives of civil entities and the public sector to participate in the meetings, but without voting power All meetings are open to citizens The committee presents a public agenda and all bulletins and opinions produced are published on the official CTNBio website Decisions by absolute majority of votes (14 votes) For the commercial release of GMOs and derivatives, the required majority is 18 votes The president of CTNBio is appointed by the MCTI Minister for a 2-year term, renewable for another 2 years All members have a two-year term, renewable for two consecutive terms The commission has a permanent executive secretariat capable of technically and administratively advising members and organizes monthly meetings (except in January and July) All decisions are published in the official journal (Diário Oficial da União) and open for public comment within 30 days Decides whether there is a need for environmental licensing for research if the GMO presents a risk to the environment or human health
CIBio	Composed of 4 members, one of whom is the chairman of the committee All members are scientists from a research institution	Members with adequate training and knowledge in the areas of Biotechnology, Recombinant DNA Technology, Biosafety and other related areas	Required for all public or private research institutions that deal with Recombinant DNA Technology and GMOs and their derivatives To operate, the commission must receive the CQB from CTNBio Registers all projects and project leaders that involve research activities with GMOs at the institution Keeps a record of monitoring all activities involving GMOs and their derivatives at the institution Responsible for ensuring the biosafety conditions of the entity’s facilities Responsible for conducting regular inspections (at least 1 inspection per year) and inspections at the research institution to attest compliance with Biosafety rules and standards Holds at least 2 annual meetings Writes and issues the annual report of its activities and projects to CTNBio Trains human resources in biosafety, conducting periodic training Takes the first appropriate measures to avoid/contain adverse effects in the event of an accident Notifies the responsible agencies in the event of accidents CIBios have the autonomy to authorize projects with GMOs and derivatives of risk class 1 Edits the institution’s Biosafety Manual
OERFs	There are 4 ministries that make up the CNBS	Ministry of Agriculture, Livestock, and Supply Ministry of Health Ministry of Environment Secretariat of Aquaculture and Fisheries	Responsible for monitoring GMOs and their by-products Inspect research activities Register and inspect the commercial use of GMOs Authorize the import of products for research and commercial use Update researchers and research institutions on Biosafety information Advise CTNBio in defining parameters for assessing biosafety Grant registrations and authorizations for commercial use of GMOs Enforce the law and apply penalties in cases of non-compliance with biosafety parameters

**Table 2 genes-16-00553-t002:** Criteria for classifying biotechnological products as GMO and non-GMO by CTNBio, according to RN16.

Criterion	Classification
(I) Absence of recombinant DNA/RNA	non-GMO
(II) Presence of genetic elements that could be obtained by crossing	non-GMO
(III) Presence of induced mutations that could also be obtained by standard techniques	non-GMO
(IV) Presence of mutations that could occur naturally	non-GMO
(V) Presence of mutations by SDN-1 *	non-GMO
(VI) Presence of mutations by SDN-2 *	non-GMO
(VII) Presence of mutations by SDN-3 *	GMO

* According to case-by-case assessments.

## Data Availability

No new data were created or analyzed in this study. Data sharing is not applicable to this article.

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
