# Peer review of "Updates on the Regulatory Framework of Edited Organisms in Brazil: A Molecular Revolution in Brazilian Agribusiness"

_genes, 2025, doi:10.3390/genes16050553_

Round 1
Reviewer 1 Report
Comments and Suggestions for Authors
da Cunha et al. is a well written and comprehensive account of the development of the regulatory framework in Brazil to accommodate “New Breeding Tools”, such as gene editing.
A few minor points were noted (presented in chronological order):
Figure 1 – “B” the legend talks about “pre-transcriptional gene silencing” when “transcriptional gene silencing” will suffice. “C” under the NHEJ image it has “loss of a few bases” should be “loss or gain of a few bases”
Line 226 – could edit “in genetic improvement…” to “via genetic improvement …”
Figure 3 – in the case of the UK the Genetic Technology (Precision Breeding) Act 2023 placed the “NBT”, termed “PBOs” by the legislation into category 3 “non-GMO/requires government authorization”.
Figure 4 – “B” intragenesis should be replaced with cisgenesis as this is what is show and described in the legend. Intragenesis has a different definition to cisgenesis. “D” is a complex diagram that was difficult to follow, could consider simplifying it (Fig 5 was much easier to understand). Also, the meaning of the various annotations (*, **, ***, gamma symbol, etc.) were not explained in the legend.
Figure 5 – “A” why not replace the black/blue asterisks with GMO/non-GMO. Also the “***” after CTNBio were not explained.
Figure 6 – The gene editing section implies that the plants are non-GMO from the start. Most will be transgenic from a period and the screening of the T0 or T1 generations will look for edited plants that have lost the transgene. More broadly, it would be useful to briefly mention this in the text and how NBTs are determined to be transgene free by the regulator.
Consider italicising in vivo and in vitro.
Does RN16 specify the extent of changes derived from SDN-1/2 approaches? Could multiple edits be included in a new elite cultivar and still pass as a NBT?
Author Response
Comments and Suggestions for Authors
da Cunha et al. is a well written and comprehensive account of the development of the regulatory framework in Brazil to accommodate “New Breeding Tools”, such as gene editing.
A few minor points were noted (presented in chronological order):
Figure 1 – “B” the legend talks about “pre-transcriptional gene silencing” when “transcriptional gene silencing” will suffice. “C” under the NHEJ image it has “loss of a few bases” should be “loss or gain of a few bases”
Response 1: Thank you for the corrections. The caption for Figure 1 has been changed as suggested by the reviewers (page 4, line 160), as has the text of Figure 1, which has been corrected.
Line 226 – could edit “in genetic improvement…” to “via genetic improvement …”
Response 2: The correction was made as suggested.
Figure 3 – in the case of the UK the Genetic Technology (Precision Breeding) Act 2023 placed the “NBT”, termed “PBOs” by the legislation into category 3 “non-GMO/requires government authorization”.
Response 3: Figure 3 has been corrected as recommended by the reviewer.
Figure 4 – “B” intragenesis should be replaced with cisgenesis as this is what is show and described in the legend. Intragenesis has a different definition to cisgenesis. “D” is a complex diagram that was difficult to follow, could consider simplifying it (Fig 5 was much easier to understand). Also, the meaning of the various annotations (*, **, ***, gamma symbol, etc.) were not explained in the legend.
Response 4: Thank you for the suggestions. Figure 4, in section "D' has been simplified and the symbols mentioned have been removed. The word "intragenesis" has been replaced by "cisgenesis", as recommended.
Figure 5 – “A” why not replace the black/blue asterisks with GMO/non-GMO. Also the “***” after CTNBio were not explained.
Response 5: Thank you for the suggestions. Figure 5 has been modified as recommended. The mentioned symbols have been removed.
Figure 6 – The gene editing section implies that the plants are non-GMO from the start. Most will be transgenic from a period and the screening of the T0 or T1 generations will look for edited plants that have lost the transgene. More broadly, it would be useful to briefly mention this in the text and how NBTs are determined to be transgene free by the regulator.
Response 6: Answer 6: Thank you for the suggestions. The explanation regarding the editing of GM plants and the loss of transgenes/cisgenes, as mandatory to be considered NBT, was added in the caption of Figure 6 (page 19, lines 462 to 464).
Consider italicizing in vivo and in vitro.
Response 7: All in vitro and in vivo were italicized.
Does RN16 specify the extent of changes derived from SDN-1/2 approaches? Could multiple edits be included in a new elite cultivar and still pass as a NBT?
Response 8: RN16 does not specify the extent of modifications made by SDN-1/2 for the plant to still be considered NBT.

Reviewer 2 Report
Comments and Suggestions for Authors
A brief summary
The rapid emerging and development of new technologies, such as genome editing in organisms, raises the necessity for establishing the regulatory mechanisms for their use. This review presents an update of the regulatory framework for genome editing activities in organisms in Brazil, which, according to published data, is in second place in terms of planted areas with GMs.
Broad comments
The review is properly designed, conducted, presented and described, and it meets the standards required for publication in a scientific journal. It contains important information that would be useful for many scientists and legislative bodies as overall, as well as for those from other countries where genome-edited organisms are used.
The abstract presents well the main idea of the review.
The keywords are four and are well selected.
The Introduction section provides information about the methods and technics for improving and obtaining desirable traits and characteristics of plants and animals, by introduce GMOs and at recent day, development and introducing a modern gene editing technology (CRISPR/Cas9) that enable to obtain with a low level of modification at non-target sites an elite edited plants in a short time and at considerably lower costs.
Other sections of the review are presented in a logical sequence. The authors cited the Brazilian Biosafety Law, which determines the role of the organizations/bodies included in the regulation of the research, monitoring and evaluation mechanisms for activities of GMOs, as well as the requests for commercial release of these organisms in Brazil.
The development of New Breeding Technologies has imposed the need to update and adjust Brazilian legislation to ensure the biosafety parameters of these biotechnological products. The authors discuss the importance of the regulatory framework that controls the exploration, manipulation, and commercial use of the edited organisms in agriculture and other sectors to ensure safety for humans, animals, and the environment in Brazil. The main instruments, systems and evaluation mechanisms used by this framework are discussed. The authors of the manuscript have highlighted the role of Normative Resolution No.16, which, on one side, allows overcoming the difficulties in the technological advancement and, on the other side, ensures safety requirements are proportional to the risks that would be determined on a case-by-case basis.
The benefits and future prospects of the Normative Resolution No.16 for edited organisms in Brazil are clearly presented in section number 5.
The conclusions are well defined.
All figures and tables are informative and well presented.
All references are appropriate.
I have only small remarks.
1. In the Introduction section, the main goal of the review should be clearly defined.
2. On line 458, it would be better to use “Adapted from Alvarez et al., [33]” instead of “Adapted from Alvarez et al., 2021 [33]”
3. Reference [46] is missing in the main text. It should be added.
Author Response
Comments and Suggestions for Authors
A brief summary
The rapid emerging and development of new technologies, such as genome editing in organisms, raises the necessity for establishing the regulatory mechanisms for their use. This review presents an update of the regulatory framework for genome editing activities in organisms in Brazil, which, according to published data, is in second place in terms of planted areas with GMs.
Broad comments
The review is properly designed, conducted, presented and described, and it meets the standards required for publication in a scientific journal. It contains important information that would be useful for many scientists and legislative bodies as overall, as well as for those from other countries where genome-edited organisms are used.
The abstract presents well the main idea of the review.
The keywords are four and are well selected.
The Introduction section provides information about the methods and technics for improving and obtaining desirable traits and characteristics of plants and animals, by introduce GMOs and at recent day, development and introducing a modern gene editing technology (CRISPR/Cas9) that enable to obtain with a low level of modification at non-target sites an elite edited plants in a short time and at considerably lower costs.
Other sections of the review are presented in a logical sequence. The authors cited the Brazilian Biosafety Law, which determines the role of the organizations/bodies included in the regulation of the research, monitoring and evaluation mechanisms for activities of GMOs, as well as the requests for commercial release of these organisms in Brazil.
The development of New Breeding Technologies has imposed the need to update and adjust Brazilian legislation to ensure the biosafety parameters of these biotechnological products. The authors discuss the importance of the regulatory framework that controls the exploration, manipulation, and commercial use of the edited organisms in agriculture and other sectors to ensure safety for humans, animals, and the environment in Brazil. The main instruments, systems and evaluation mechanisms used by this framework are discussed. The authors of the manuscript have highlighted the role of Normative Resolution No.16, which, on one side, allows overcoming the difficulties in the technological advancement and, on the other side, ensures safety requirements are proportional to the risks that would be determined on a case-by-case basis.
The benefits and future prospects of the Normative Resolution No.16 for edited organisms in Brazil are clearly presented in section number 5.
The conclusions are well defined.
All figures and tables are informative and well presented.
All references are appropriate.
I have only small remarks.
- In the Introduction section, the main goal of the review should be clearly defined.
Response 1: Thank you for the suggestion. At the end of the Introduction, the main purpose of the review was added to the text (Page 5, lines 175 to 178).
- On line 458, it would be better to use “Adapted from Alvarez et al., [33]” instead of “Adapted from Alvarez et al., 2021 [33]”
Response 2: Thank you for the suggestion: The excerpt was modified according to the reviewer's recommendation (page 19, line 465).
- Reference [46] is missing in the main text. It should be added.
Response 3: Answer 3: Reference 46 is found on page 18, line 446.
